# Oral Lichen Planus and Dental Implants: Protocol and Systematic Review

**DOI:** 10.3390/jcm9124127

**Published:** 2020-12-21

**Authors:** Aina Torrejon-Moya, Constanza Saka-Herrán, Keila Izquierdo-Gómez, Antoni Marí-Roig, Albert Estrugo-Devesa, José López-López

**Affiliations:** 1Department of Oral Medicine, Surgery and Implantology, Faculty of Medicine and Health Sciences (Dentistry), University of Barcelona, 08907 Barcelona, Spain; aina.torrejon@gmail.com; 2Faculty of Medicine and Health Sciences (Dentistry), University of Barcelona, 08907 Barcelona, Spain; constanzasakah@gmail.com; 3Oral Health and Masticatory System Group (Bellvitge Biomedical Research Institute), IDIBELL, Department of Odontostomatology, Faculty of Medicine and Health Sciences (Dentistry), University of Barcelona, 08907 Barcelona, Spain; keila_izqdo@hotmail.com (K.I.-G.); ebusitano@gmail.com (A.M.-R.); albertestrugodevesa@gmail.com (A.E.-D.); 4Department of Maxillofacial Surgery, University Hospital of Bellvitge, 08907 Barcelona, Spain; 5Department of Odontostomatology, Faculty of Medicine and Health Sciences (Dentistry), Odontological Hospital University of Barcelona, University of Barcelona, 08907 Barcelona, Spain

**Keywords:** oral lichen planus, dental implants, survival rate

## Abstract

A systematic review was conducted to answer the following PICO question: “Can patients diagnosed with oral lichen planus (OLP) be rehabilitated with dental implants as successfully as patients without OLP?”. A systematic review of the literature was done following the Preferred Reporting Items for Systematic Reviews and Meta-Analyses (PRISMA) statements to gather available and current evidence of oral lichen planus and its relationship with dental implants. The synthesis of results was performed using a Binary Random-Effects Model meta-analysis. Summary measures were odds ratios (ORs), frequencies, and percentages comparing the survival rate of dental implants placed in patients with OLP vs. those in patients without OLP. The electronic search yielded 25 articles, after removing the duplicated ones, 24 articles were selected. Out of the 24 articles, only 15 fulfilled the inclusion criteria. According to the results of the meta-analysis, with a total sample of 48 patients with OLP and 49 patients without OLP, an odds ratio of 2.48 (95% CI 0.34–18.1) was established, with an I^2^ value of 0%. According to the Strength of Recommendation Taxonomy (SORT) criteria, level A can be established to conclude that patients with OLP can be rehabilitated with dental implants.

## 1. Introduction

Oral lichen planus (OLP) is an autoimmune and chronic inflammatory disease [1,2,3]. The etiology of OLP is still unknown, but it is believed to be associated with a cell-mediated immune dysregulation [4]. 

The prevalence of this inflammatory disease involves 1–2% of the population, the epidemiology od OLP was evaluated in 1987 by Axell and Rundquist [5], on this occasion it was concluded that 16% of the cases were among men and 22% among women. The highest prevalence was found in patients between the age of 55 and 74 years. 

Therefore, OLP usually appears in women and in the age group of 50 to 70 years old. 

The meta-analysis carried out by Aghbari et al. [2] concluded that 1.1% of patients with OLP developed oral squamous cell carcinoma (OSCC). 

Since OLP is a mucosal disease, it has been suggested to affect the ability of attachment to the titanium surface; and although some conditions are considered as risk factors for dental implants, there are only a few absolute contraindications for this rehabilitation option [1]. 

Taking this statement into account, oral rehabilitation with dental implants in patients with OLP needs to be questioned; in fact, surgical injury due to the implant insertion procedure should be avoided during active, erosive phases of OLP. Frequent follow-ups are needed in order to rule out inflammatory tissue response interfering with long-term survival of implants [1]. 

Excellent results in patients without general pathologies have been reported with survival and success rates of implant, in a long-term follow-up [6]. Furthermore, oral rehabilitation with dental implants has a 92–95% success rate in patients without any oral or systemic condition [7].

However, other factors such as smoking and the level of oral hygiene can affect osseointegration and lower dental implant success rates [8]. 

The main aim when treating patients with OLP is to avoid irritating factors for the mucosa, since this disease may appear in either asymptomatic or symptomatic forms. Patients may refer to burning or occasional pain [1,2,3,4].

This systematic review was conducted to answer the following PICO question: “Can patients diagnosed with OLP be rehabilitated with dental implants with the same survival rate as patients without OLP?”. 

## 2. Experimental Section

A systematic review of the literature was done following the Preferred Reporting Items for Systematic Reviews and Meta-Analyses (PRISMA) statements [9] to gather available and current evidence of oral lichen planus and its relationship with dental implants (Figure 1). The review was carried out from February 2020 to April 2020. Electronic research without restriction dates was carried out using two different electronic databases: PubMed and the Cochrane Central Register for Controlled Trials.

The following terms were searched in PubMed and Cochrane; (“Lichen Planus, Oral” [Mesh]) AND “Dental Implants” [Mesh]. 

Inclusion criteria were articles written in English or Spanish, meta-analyses, systematic reviews, randomized-control trials, cohort studies, and case reports. 

On the other hand, exclusion criteria were animal studies, in vitro studies, and descriptive reviews.

The primary outcome of this article was to establish whether patients with OLP had the same survival rate as patients without OLP. Furthermore, secondary outcomes such as the protocol and recommendations for patients with OLP who were rehabilitated with dental implants were reviewed. 

The following data were extracted from the included studies (when available) by two independent reviewer authors (A.T.-M. and C.S.-H.): authors, year, study design, number of subjects, gender, age, OLP type, OLP location, OLP duration (in years), previous biopsy, number of implants, type of prostheses, survival rate, follow-up (in months), malignant transformation, and OLP treatment. Discrepancies were resolved with the other authors (K.I.-G., A.M.R., and A.E.-D.), and finally data were validated by J.L.L.

The selected studies were assessed following the Strength of Recommendation Taxonomy (SORT) criteria [10].

A quality assessment of the publications included for the systematic review was not applicable due to the entity of included publications. Blinding of participants and personnel (performance bias) and blinding of outcome assessment (detection bias) were not applicable, since most publications were case reports. Due to selective reporting within case reports, attrition bias (incomplete outcome data) and reporting bias (selective reporting) were assumed to be high.

### Statistical Analysis

The synthesis of results was performed using a Binary Random-Effects Model meta-analysis. Summary measures were odds ratios (ORs), frequencies, and percentages comparing the survival rate of dental implants placed in patients with OLP vs. those in patients without OLP. 

Forest plots were produced to graphically represent the odds ratio of successful placement of the dental implant in patients with OLP and among patients without OLP (Figure 2). 

Heterogeneity was assessed by the I^2^ statistics. Each outcome was combined and calculated using the Review Manager software. 

## 3. Results

The electronic search, in PubMed and Cochrane databases, yielded 25 articles, after removing the duplicated ones, 24 articles were selected. Out of the 24 articles, only 15 fulfilled the inclusion criteria. 

As shown in Table 1, all articles [11,12,13,14,15,16,17,18,19,20,21,22,23,24] were level 2, according to the SORT criteria [10], except for a retrospective study [15] and a prospective controlled trial [25] that were level 1. Most of the articles [11,12,13,14,15,16,17,18,19,21,24] were case reports, there were three retrospective studies [15,20,23] and two prospective studies [22,25]. 

A total of 110 patients were examined, with an age range of 51 to 81 years old. There were a total of 83 female patients (75.45%) and 27 male patients (24.55%). Not all the studies evaluated the type of OLP, although a total of 72 patients were divided by the type of OLP. The most common OLP type was erosive OLP (42 cases—58.33%), followed by the reticular OLP (22 cases—30.55%) and the atrophic type (8 cases—11.11%). Regarding the location of OLP, a total of 29 locations were mentioned; the most common location was the buccal mucosa (22 cases—75.86%), following the gingiva (5 cases—17.24%), the tongue (1 case—3.44%), and the palate (1 case—3.44%). 

From the 15 articles selected, only 8 [12,16,17,19,20,23,24,25] mentioned realizing a biopsy to diagnose OLP, and the survival rate was only mentioned in 9 articles [12,13,14,17,18,20,21,22,23,25], the mean of the survival rate was 93.88%.

Regarding the type of prostheses, it was analyzed in most of the articles, and a total of 113 prostheses were analyzed, the most used type of protheses was the fixed partial (95 rehabilitations—84.07%), the second most used was the fixed complete rehabilitation (10 rehabilitations–8.84%), and the overdenture (8 rehabilitations–7.07%). 

Concerning OLP treatment when patients were being rehabilitated with dental implants, it was only considered in 6 articles [17,20,22,23,24,25]. All of them used oral corticosteroids with different active ingredients and different posology. The most used oral corticosteroid was triamcinolone acetonide 0.01 3 times/day [20,24], except for Hernández et al. [25], who used prednisone 30 mg 1 time/day for 5 to 10 days in 2 patients, and Anitua et al. [23], who used deflazacort 20 mg 2 days preoperative and 15 mg postoperative during 3 days and 7.5 mg during 3 days. Czerninski et al. [17] used dexamethasone 0.4%, triamcinolone 8 mg, or clobetasol propionate ointment 0.05% applied once or twice daily for not more than 2 weeks as an initial treatment, but during the follow-up, they changed to less-potent steroids. Furthermore, Aboushelib and Elsafi [22] used a diode laser as an alternative therapy. 

Regarding malignant transformation, 5 articles stated a malignant transformation [15,16,18,19,20]. A total of 8 patients (7.3%) were diagnosed with a malignant pathology after placing the implants. Of these, 2 patients (25%) had risk factors such as alcohol consumption and smoking habits, 4 patients (50%) did not have any kind of risk factor, and in the case of the 2 remaining (25%) patients it was not reported. 

Regarding the 8 patients who were diagnosed with oral squamous cell carcinoma, only 4 of them [16,19] had a previous biopsy confirming the oral lichen planus; therefore, only 50% of the patients presented malignant transformation had a previous histopathological confirmation of oral lichen planus.

Moreover, 3 patients (37.5%) reported a previous malignant pathology, 3 patients (37.5%) reported not having any history of OSCC, and for the 2 remaining patients (25%) it was not reported. 

According to the results of the meta-analysis, with a total sample of 48 patients with OLP and 49 patients without OLP, an odds ratio of 2.48 (95% CI 0.34–18.1) was established. Indicating that patients with OLP had a 2.48 times higher survival with dental implants in comparison with patients without OLP, and the I^2^ was used to assess heterogeneity between studies.

## 4. Discussion

Based on our systematic review of the literature and answering our PICO question: rehabilitation with dental implants in patients with OLP is a valid treatment option with a survival rate of 93.88%. This survival rate is similar to the survival rate of implant rehabilitation in patients without any pathology or systemic condition, which is 92% or 95%, depending on the prosthetic rehabilitation [7]. 

In order to maintain this high survival rate, frequent follow-up appointments and oral hygiene instructions should be established to eliminate inflammatory tissue response (peri-implant mucositis and peri-implantitis) [1].

Although the most common type of OLP is the reticular type [26], followed by the erosive type, in this review, the most common type was the erosive OLP, this could be explained because not all studies reported which type of OLP was diagnosed or because some professionals ignore or do not think the reticular type is relevant. The buccal mucosa was the most common location for OLP in agreement with other articles [27,28]. 

We would like to point out, that we were surprised by the fact that none of the articles mentioned previously reported any case of re-activation of OLP or an atrophic or erosive stage after the placement of dental implants. 

Furthermore, although the malignant transformation rate calculated (7.3%) is substantially higher than the one reported by Aghbari et al. [2] in patients with OLP, 50% of the registered cases were not confirmed with a previous biopsy, therefore we cannot confirm that they were OLP cases. As well, we should consider that these results are not extendable to a general population since the patient sample reported is very small. 

Additionally, we were surprised by the fact, that not all the studies that reported malignant transformation, reported the time that elapsed between the OLP diagnoses and the development of OSCC, and again, the frequency with which the patients were evaluated was not mentioned. 

Another notable matter is the fact that only 53.33% of the studies analyzed related performing a biopsy to diagnose the OLP. However, both clinical and histopathologic criteria need to be present to diagnose OLP, according to the modified WHO diagnostic criteria of OLP and oral lichenoid lesions, proposed by van der Meij and van der Waal [29]. Therefore, we consider that it is difficult to know if the other 46.67% were correctly diagnosed since there was no reference to histopathologic evaluation. 

Regarding the meta-analysis, an odds ratio of 2.48 was established, indicating that patients with OLP have a higher survival rate in comparison with patients without OLP. However, these results are based on only 2 studies [20,25], since they were the only studies with patients with OLP and a control group in the literature. These results may be explained since patients with OLP have more frequent controls and a more meticulous oral hygiene control, therefore any alteration could be detected earlier. These results should be evaluated with caution since there are only two studies and with small sample sizes. We recommend new studies with larger sample size and longer follow-up. 

According to our results, implant rehabilitation in patients with OLP is not contraindicated, and survival rates are comparable to those of patients without OLP. 

Given the foregoing [30], our workgroup would like to propose a protocol for OLP treatment and monitoring when rehabilitated with dental implants (Figure 3).

(i) Initially, we would like to remark on the importance of a detailed medical history, a photographic register, and a biopsy to confirm the OLP diagnosis. 

(ii) Secondly, and in agreement with Hernández et al. [25], no implants should be placed until the remission of atrophic or erosive forms is achieved, in other words, no patients should be treated during a flare-up period of the disease. Erosive and atrophic OLP is mostly symptomatic and pain can be one of the main characteristics. Therefore, if we are to rehabilitate a patient with symptomatic OLP, we agree with Aboushelib and Elsafi [22] and Anitua et al. [23] and propose a prophylactic corticosteroid therapy, to avoid the reactivation of erosive and atrophic OLP after the implant surgery. Again, other authors, such as Hernández et al. [25] also reported using systemic corticosteroids to manage acute phases of OLP. 

The guideline that we propose is deflazacort 30 mg 2 days pre-operative, 15 mg 3 days post-operative, and 7.5 mg for 3 more days. We propose deflazacort since it is a synthetic glucocorticoid characterized by high efficacy and good tolerability, widely used in autoimmune disease and dermatology [31]. 

From here, we agree with other studies [20,24] and would recommend mouthwashes of 0.01% triamcinolone acetonide three times per day, until the remission of the acute forms. 

(iii) Finally, meticulous oral hygiene and frequent, regular appointments [32] are important to prevent inflammatory tissue response, such as mucositis or peri-implantitis, and for the early detection of malignant transformation [33,34,35].

The limitations of this systematic review include the fact that most studies were case reports, which means low-level evidence, and the fact that case series did not provide data separately. Additionally, the published cases provided a short follow-up, therefore longer follow-ups would mean a more reliable prognosis.

Therefore, with the discrepancies presented in the current literature along with a lack of treatment guidelines and the small sample analyzed, caution should be exercised when drawing a conclusive statement.

## 5. Conclusions

According to the SORT criteria [9], level A can be established to conclude that patients with OLP can be rehabilitated with dental implants. 

More studies with longer follow-up and higher levels of evidence, such as randomized-controlled trials, are needed.

## Figures and Tables

**Figure 1 jcm-09-04127-f001:**
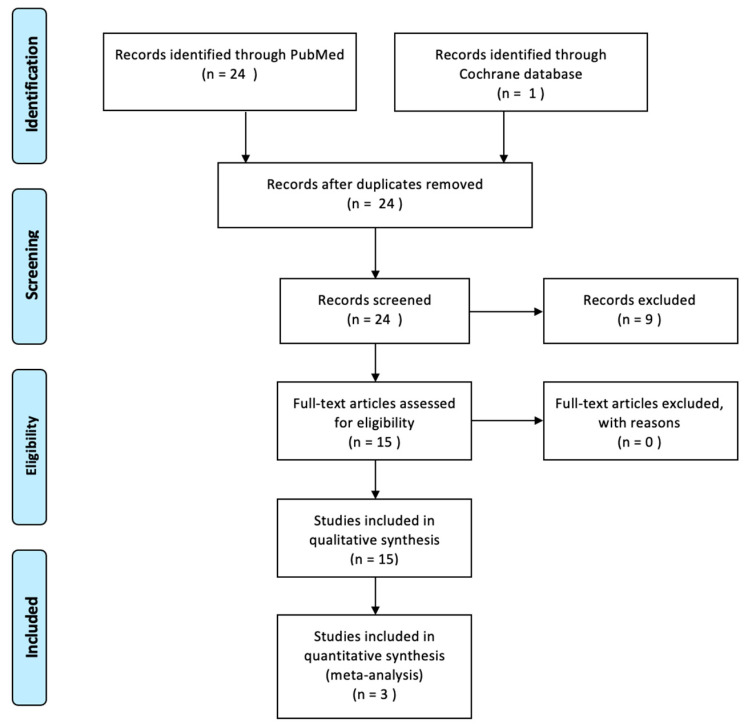
Preferred Reporting Items for Systematic Reviews and Meta-Analyses (PRISMA) flow diagram.

**Figure 2 jcm-09-04127-f002:**
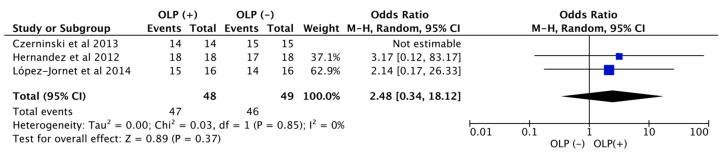
Meta-analysis forest plot. OLP, oral lichen planus.

**Figure 3 jcm-09-04127-f003:**
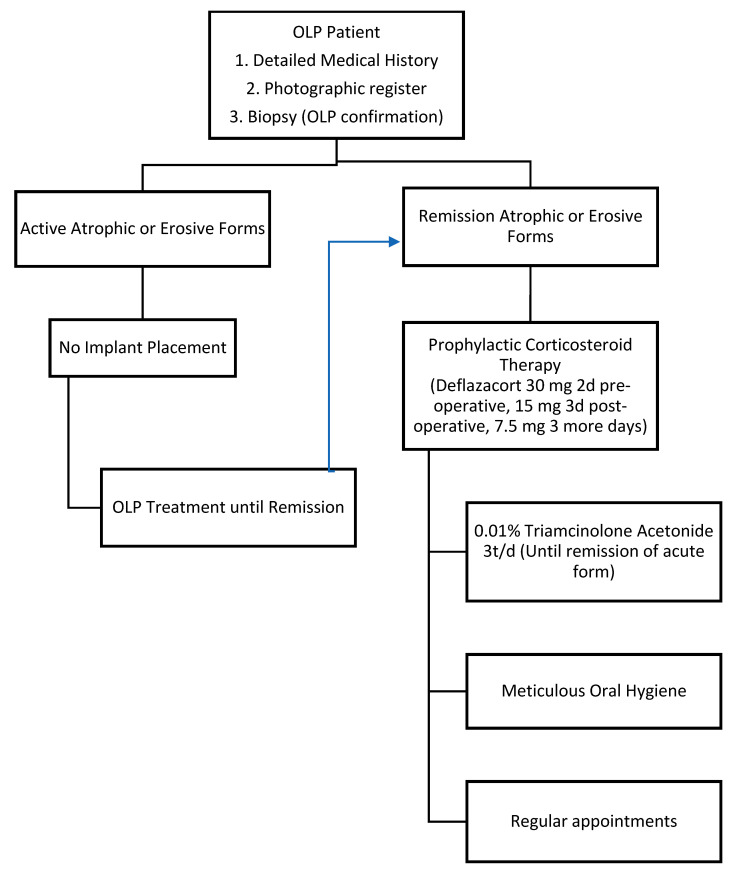
Protocol proposed by our workgroup.

**Table 1 jcm-09-04127-t001:** Summary of the studies evaluated.

Article	Study Design	Sort	Nº Patients	Gender	Age	OLP Type	OLP Location	OLP Duration (Years)	OLP Biopsy	N° Implants	Protheses	Survival Rate	Follow-up (Months)	OSCC	OLP Treatment
Esposito et al. (2003)	Cr	2	1	F	69	Erosive	-	-	Ni	2	Overdenture	-	32/60		-
Esposito et al. (2003)	Cr	2	2	F	72 78	Erosive	Buccal mucosa Gingiva	16	Yes	4	Overdenture	100%	21		-
Öczakir et al. (2005)	Cr	2	1	F	74	-	-	-	Ni	4	Fixed complete	100%	72		
Reichart (2006)	Cr	2	3	F	63 68 79	Reticular Atrophic	Buccal mucosa, Gingiva	10 12 20	Ni	10	Fixed partial	100%	36		-
Czerninski (2006)	Cr	2	1	F	52	Erosive		8	Ni	3	Fixed partial	-	36	Yes	
Gallego et al. (2008)	Cr	2	1	F	81	Reticular	Buccal mucosa, tongue, palate	-	Yes	2	Overdenture	-	36	Yes	-
Hernandez et al. (2012)	Ps	1	18	14F 4M	53.5 (M)	Erosive	Buccal mucosa Gingiva	-	Yes	56	Fixed partial	100%	53.5		Clobetasol propionate 0.05% 3t/d (18 patients) + prednisone 30 mg 1t/d 5-10d (2 patients)
Czerninski et al. (2013)	Rs	2	14	11F 3M	59.5 (M)	Erosive Atrophic Reticular	Buccal mucosa, gingiva	-	Yes	54	-	100%	12–24		Dexamethasone 0.4%/triamcinolone 8 mg or clobetasol propionate 0.05% 1-2t/d no more than 2 weeks
Marini et al. (2013)	Cr	2	1	F	51	Reticular			Ni	2	Fixed partial	50%	108	Yes	
Moergel et al. (2014)	Cr	2	3	F	54 69 80	-			Yes				6–51	Yes	
López-jornet et al. (2014)	Rs	2	16	10F 6M	64.5 (M)	Erosive Reticular		-	Yes	56	Overdenture Fixed partial	96.42%	42		0.01% triamcinolone acetonide 3t/d
Raiser et al. (2016)	Cr	2	2	F	55 70	-	-	-	Ni	10,6	Fixed complete, Fixed partial	100%	96, 36	Yes	-
Aboushelib et al. (2017)	Ps	2	23	12F 11M	56.7 (M)	-	-	-	Ni	55		-	-		Oral corticosteroids + diode laser
Anitua et al. (2018)	Rs	1	23	20F 3M	58 (M)	Erosive Reticular			Yes	66	Fixed partial Fixed complete	98.5%	68		Deflazacort 30 mg 2d preoperatively, 15 mg postoperatively 3d and 7.5 mg 3d
Fu l et al. (2019)	Cr	2	1	F	65	Erosive		5	Yes	4	Overdenture		36		0.01% triamcinolone acetonide 3t/d

OSCC: oral squamous cell carcinoma, OLP: oral lichen planus Cr: case report, Ps: prospective study, Rs: retrospective study, M: mean, Ni: not included.

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
