# Peer review of "Oral Lichen Planus and Dental Implants: Protocol and Systematic Review"

_jcm, 2020, doi:10.3390/jcm9124127_

Round 1
Reviewer 1 Report
Dear Authors
the paper is interesting
however, you should include in your introduction and discussion the following reference about implants in patients at risk for oral lichen planus
- PubMed ID: 26238779
- PubMed ID: 25955953
Author Response
Dear Reviewer,
Thank you very much for your reply.
We have read the two articles that you proposed, and although they are about HIV patients, we have added some of their considerations to our article, as you will see in references 8 and 33, both introduction and discussion. (Line 63) (Line 216)
“Although, other factors as smoking and the level of oral hygiene can affect osseointegration and lower dental implants success. [8]”
“Finally, meticulous oral hygiene and frequent regular appointments [32] are important to prevent inflammatory tissue response, such as mucositis or peri-implantitis, and for early detection of malignant transformation [33-35].”
Kind Regards,
Aina Torrejon Moya
Reviewer 2 Report
This is a good article, but in my opinion it cannot be considered a systematic review. I would combine the study in literature review.
Author Response
Dear Reviewer,
Thank you very much for your reply.
We know there’s few articles and the sample is small. Concerning this comment, you will find we have extended our limitations paragraph (line 227).
“Therefore, with the discrepancies presented in the current literature along with a lack of treatment guidelines, and the small sample analyzed, caution should be exercised when drawing a conclusive statement.”
Kind Regards,
Aina Torrejon Moya
Reviewer 3 Report
First of all I would like to thank you for the efforts placed to conduct this systematic review. The work has an interesting topic: the insertion of implants in patients with oral lichen planus.
My main concerns are the following:
Introduction
In this section it was not clearly explained why lichen could affect the osseointegration process.
Experimental section
- Lines 78-79: Inclusion and exclusion criteria are not clear, please explain well.
- Please describe primary and secondary outcomes.
- Did You consider only survival rate or success rate too?
Statistical analysis
Sample size was too small to conduct a meta-analysis (only 2 articles were included)
Results
- The results of systematic review were presented exhaustively.
- It is necessary a section that describe potential risk of bias of included studies.
- Both clinical and histopathologic criteria need to be present to diagnosticate an OLP, why did You decide to include the articles in which biopsy was not done? please consider to include only the articles where OLP was diagnosticated with a biopsy.
Discussion
The discussion is off topic because it focused mainly on the malignant transformation of lichen planus (lines 155-170). The discussion needs to be reorganised and structured and the relationship between implant success and lichen should be further explored.
In conclusion, please eventually consider to eliminate the meta-analysis, presenting only the results of the systematic review.
Author Response
Dear Reviewer,
Thank you very much for your reply.
Below, we answer all the questions that the reviewer indicated.
Introduction
“In this section it was not clearly explained why lichen could affect the osseointegration process.”
# As you will see, an explanation has been added to the introduction. (Line 58)
“Taking this statement into account, oral rehabilitation with dental implants in patients with OLP needs to be questioned; considering dental implants penetrate the oral mucosa, and it has been suggested that OLP may directly affect the nature of the barrier effect of the mucosa. [1]”
Experimental section
- Lines 78-79: Inclusion and exclusion criteria are not clear, please explain well.
#This has been clarified as you will find in line 82 – 84
“Inclusion criteria were articles written in English or Spanish, Meta-analysis, Systematic Reviews, Randomized control trials, Cohort studies, and Case reports.
On the other hand, exclusion criteria were animal studies, in vitro studies, and reviews.”
- Please describe primary and secondary outcomes.
# Primary and Secondary Outcomes are now described in line 85.
“The primary outcome of this article was to establish if OLP patients had the same survival rate as non-OLP patients. Furthermore, secondary outcomes such as protocol and recommendations for OLP patient rehabilitated with dental implants were reviewed. “
- Did You consider only survival rate or success rate too?
# Only Survival Rate was considered. Any disarranging regarding this matter has been reviewed and clarified.
Statistical analysis
Sample size was too small to conduct a meta-analysis (only 2 articles were included)
# An statement has been added to the limitations paragraph (Line 227) in order to alert the importance of drawing conclusive statements of the article, and it is also described in line 194.
“These results should be evaluated with caution since there are only two studies and with small sample size. We recommend new studies with larger sample size and longer follow-up.”
“Therefore, with the discrepancies presented in the current literature along with a lack of treatment guidelines, and the small sample analyzed, caution should be exercised when drawing a conclusive statement.”
Results
- The results of systematic review were presented exhaustively.
# Thank you very much for your consideration.
- It is necessary a section that describe potential risk of bias of included studies.
# The potential risk of bias is now described in line 95.
“A quality assessment of the publications included for the systematic review was not applicable due to the entity of included publications. Blinding of participants and personnel (performance bias), blinding of outcome assessment (detection bias) were not applicable, since most publications were case reports. Due to selective reporting within case reports, attrition bias (incomplete outcome data), reporting bias (selective reporting) were supposed high.”
- Both clinical and histopathologic criteria need to be present to diagnosticate an OLP, why did You decide to include the articles in which biopsy was not done? please consider to include only the articles where OLP was diagnosticated with a biopsy.
# Articles were included because it wasn’t indicated that biopsies weren’t done, it simply was not included in the article, so we don’t know if they were or they weren’t carried out. Therefore, we accepted the clinicals diagnoses of OLP although, it wasn’t reported if the biopsy was made.
Discussion
The discussion is off topic because it focused mainly on the malignant transformation of lichen planus (lines 155-170). The discussion needs to be reorganised and structured and the relationship between implant success and lichen should be further explored.
# The malignant transformation paragraph has been deleted and the discussion has been reorganized. Furthermore, more importance to the survival rate is now given (Line 164).
“In order to maintain this high survival rate, frequent follow-up appointments and oral hygiene instructions should be established to eliminate inflammatory tissue response (peri-implant mucositis and peri-implantitis). [|]”
In conclusion, please eventually consider to eliminate the meta-analysis, presenting only the results of the systematic review.
# Although there are only 2 articles, we have described this limitation in the discussion of our article, and it should be noted to the readers that there’s only two articles and this should encourage more authors to publish articles with larger samples.
Reviewer 4 Report
The manuscript describes methodological studies that provide useful information on the plausible success of oral rehabilitation with dental implants in patients with oral lichen planus.
Although several studies show favorable outcomes following the application of dental implants in OLP patients, it is impossible to understand where the consensus view stands, especially because insufficient studies are only available in the current literature. The authors did not sufficiently discuss the claims for some factors that caused implant failure in OLP patients as mentioned in ref 10 of the manuscript. With these discrepancies in the current literature along with the lack of treatment guidelines, caution should be exercised when drawing a conclusive statement. The authors need to discuss more on articles that failed in implant therapy as well in order to provide a balanced view of the analysis. The authors need to emphasize the need for controlled studies in order to provide an accurate assessment of the merits and demerits of applying dental implants in OLP patients.
Author Response
Dear Reviewer,
Thank you very much for your reply.
I would like to point out, that this statement has been added, as you will see in the discussion – in the limitations part (line 227) -, pointing out that caution should be exercised when drawing a conclusive statement with all the present discrepancies.
“Therefore, with the discrepancies presented in the current literature along with a lack of treatment guidelines, and the small sample analyzed, caution should be exercised when drawing a conclusive statement.”
Kind Regards,
Aina Torrejon Moya
Round 2
Reviewer 3 Report
Dear Authors,
Thank You for your response. I appreciate You followed my suggestions to correct the manuscript.
Please consider following minor revisions:
- in line 58-59 We suggest to cancel the text " considering dental...effect of the mucosa", and write: in fact the surgical injury due to the implant insertion procedure should be avoided during active, erosive phases of OLP. Frequent follow up are needed in order to rule out inflammatory tissue response interfering with long-term survival of implants.
- in line 85 please correct reviews in "descriptive reviews"
Thank You
Best regards
Author Response
Dear Reviewer,
Thank you very much for your reply!
Both changes weere made, as you will see in line 58 ("Taking this statement into account, oral rehabilitation with dental implants in patients with OLP needs to be questioned; in fact the surgical injury due to the implant insertion procedure should be avoided during active, erosive phases of OLP. Frequent follow up are needed in order to rule out inflammatory tissue response interfering with long-term survival of implants") and line 85 (On the other hand, exclusion criteria were animal studies, in vitro studies, and descriptive reviews).
Thank you very much
Kind regards
Aina Torrejon Moya